# Development and Validation of the Weighted Index for Childhood Adverse Conditions (WICAC)

**DOI:** 10.3390/ijerph192013251

**Published:** 2022-10-14

**Authors:** Sofie A. Jacobsen, Bo M. Bibby, Lisbeth Frostholm, Marie W. Petersen, Eva Ørnbøl, Signe U. Schovsbo, Thomas M. Dantoft, Tina B. W. Carstensen

**Affiliations:** 1Research Clinic for Functional Disorders and Psychosomatics, Aarhus University Hospital, Palle Juul-Jensen Boulevard 11, 8200 Aarhus, Denmark; 2Department of Clinical Medicine, Aarhus University, Palle Juul-Jensen Boulevard 11, 8200 Aarhus, Denmark; 3Department of Public Health, Biostatistics, Aarhus University, Bartholins Allé 2, 8000 Aarhus, Denmark; 4Center for Clinical Research and Disease Prevention, Bispebjerg and Frederiksberg Hospital, The Capital Region, 2400 Copenhagen, Denmark

**Keywords:** index measurement, validation, development of weighted index, hypothesis testing, adverse childhood experiences, social vulnerability

## Abstract

Background: Adverse experiences in childhood are a major public health concern, promoting social inequality in health through biopsychosocial mechanisms. So far, no known measures comprehend the complexity and variations of severity of adverse events. This study aims to develop and validate a new index: the Weighted Index for Childhood Adverse Conditions (WICAC). Methods: The population consists of 7493 randomly invited men and women aged 18–72 years. Data were collected in 2012–2015 as part of the Danish Study of Functional Disorders (DanFunD). Content and construct validation of the WICAC was performed with the hypothesis testing of multiple biopsychosocial outcomes: cardiovascular disease, cancer, poor health, back pain, BMI, obesity, anxiety, depression, low vitality, subjective social status, lower education, smoking, and alcohol consumption. Data were analysed with binominal and linear regression models with risk ratios (RR) and mean differences (MD). Results: Content validation is fitting for WICAC. The strongest associations observed were for most severe adversity: Poor Health RR = 2.16 (1.19–2.91), Anxiety RR = 3.32 (2.32–4.74), Heavy Drinking RR = 4.09 (1.85–9.04), and Subjective Social Status MD = −0.481 (−0.721–(−0.241)). Similar results were found for the remaining outcomes. Discriminative validation was undecided. Conclusions: WICAC is an adequate instrument for measuring cumulative adverse life events in childhood and adolescence for research purposes.

## 1. Introduction

Social inequity and inequality are known to be large public health concerns [1]. Many factors affect the social status of individuals, including exposure to adverse life events. It has been argued that adverse childhood experiences can affect a range of biopsychosocial aspects of adulthood. Studies have shown associations between a traumatic childhood or adverse childhood experiences (ACE) and overall poor health [2,3,4] as well as increased mortality [5]. Investigations have specifically examined adverse somatic health outcomes, such as chronic diseases, cancer, cardiovascular diseases, respiratory diseases, and diabetes [2,4,6,7,8], and poor health outcomes, such as pain conditions [2,9] and overweight [6,7,10]. As regards mental health, the known consequences of ACEs are anxiety, depression [4,6,7,8], schizophrenia, eating disorders, and suicide [2,11,12], as well as alcohol abuse, smoking, drug use [13], and risky sexual behaviour [2]. Furthermore, ACEs increase the risk of being unemployed [14] and having a lower socioeconomic position in adulthood [2,15]. Exposure to a single adverse experience rarely seems to cause long-term harm. In contrast, experiencing multiple adverse events often have lifelong health consequences [16,17] as traumatic experiences seem to have a cumulative effect [1].

A variety of different instruments to measure ACEs exist [18]. The most frequently used and known measure is the Adverse Childhood Experience questionnaire (ACE-Q), which is often used in a short form with 10 questions [3]. However, the ACE-Q has been criticized for lacking of a rational selection of the adversities constituting the construct, and severity, frequency, and duration of ACEs are not considered [19]. In addition, the ACE-Q lacks structural and social variables, such as witnessing violence in your community, poverty, and separation from family [8,20,21,22]. In response to this critique, studies have made their own modifications when measuring adversities using the ACE-Q. As the majority of measures are made for clinical purposes and qualitative interviewing [18], they are often too complex to use in larger quantitative studies or contain too many and too specific questions. They often solely focus on either psychological or social events, and only few include both childhood and adolescence experiences. As ACEs cover a wide range of adversities range from parents’ divorce to sexual abuse, it is highly important to comprehend the complexity of the variation in adversities by weighing the adversities according to the impact they may have on each other as well as their severity, repetition, and duration [19,23]. Existing measures that weigh different adversities are, however, generally sparse and field specific, and they are either made for clinical interviews [24], psychological screenings [25,26], or measurements in children [27].

The Cumulative Lifetime Adversity Measurement (CLAM) [28,29] is an unweighted retrospective measure for adults that measures lifetime adversity. The CLAM includes a variety of different biopsychosocial aspects. CLAM examines the cumulative adversity effect of a range of events or the same event if it happened more than once during a person’s lifetime. However, although it is easy to obtain data for a period by excluding adverse experiences from ages above 18 years, CLAM was not specifically developed for measuring adversities in childhood and adolescence.

As a result of this inconsistency across the variety of measures, the development of a more adaptable, accurate, and transparent instrument for measuring ACEs has been pursued [19,22,23,30].

AIM: This study aimed to develop and validate a new index for measuring cumulative adversities from ages 0–18 years, the Weighted Index for Childhood Adverse Conditions (WICAC) with CLAM as the launch pad. Furthermore, we aimed to evaluate the construct validity by hypothesis testing based on evidence in the literature for associations between both somatic, psychological, and social aspects.

## 2. Materials and Methods

### 2.1. Population Sample and Ethics

The study sample was part of The Danish study of Functional Disorders (DanFunD) [31]. DanFunD is a population-based cohort investigating the epidemiology of functional somatic disorders. It includes data from screening questionnaires for functional somatic disorders, health-related questionnaires, a general health examination, etc. Data on adverse experiences were collected using the CLAM [28,29] between 2012 and 2015. For the DanFunD study, Part Two, a total of 25,368 people were invited, of whom 7493 participated to complete the CLAM questionnaire with full data available for 6360 participants (see Figure 1). Participants were men and women aged 18–72 years living in the western part of greater Copenhagen. Exclusion criteria were not born in Denmark, not being a Danish citizen, and pregnancy. All participants completed a questionnaire, underwent a clinical examination, and gave written informed consent at the Center for Clinical Research and Prevention (formerly The Research Centre for Prevention and Health), Glostrup, Denmark. The study was approved by the Ethical Committee of Copenhagen County (H-3-2012-0015) and the Danish Data Protection Agency (2012-58-006, 1-16-02-227-16) and was conducted in accordance with the Helsinki II Declaration [28].

### 2.2. Development of the WICAC Index

For the development of the index, the guidelines by Vet et al. were used: 1. definition and elaboration of the construct intended to be measured, 2. choice of measurement method, 3. selecting and formulating items, and 4. scoring issues [32]. Pilot testing had already been conducted on the CLAM [28]. As all items in the WICAC were included in the CLAM, there was no need for further pilot testing of the WICAC.

#### 2.2.1. Definition and Elaboration of the Construct

To develop the construct, a systematic approach to a literature search was conducted [33]. The search was divided into two strategies. The first search included translations, the use of external experts, librarians, and MesH term searches aiming to define the terms of the construct. The second search aimed to investigate and search for tools and questionnaires in the field as well as to find associations between biopsychosocial perspectives and measures of adverse childhood experiences to develop our hypotheses (Appendix A: Flow chart).

#### 2.2.2. Selecting and Scoring Items

The aspects were selected theoretically according to similar measurement instruments [3,25,26,34,35,36,37,38,39,40] (Appendix A) and reflections on possible items. The item selection was, however, restricted to the items of the CLAM questionnaire [28].

Each item was scored with a weight between one and three to give higher weight to more adverse experiences and lower weight to less adverse experiences. A combined empirical and judgmental method was used with the inclusion of relevant theories for weighting each item. This was carried out in collaboration with an expert panel consisting of one child psychiatrist, one adult psychiatrist, two psychologists, one social worker, and one public health professional. Each expert was given instructions and weighted all items. Hereafter, each weight was discussed at a joined meeting. The weighing of items was based on three basic hierarchical principles:items indicating more adverse experiences were rated upwards, i.e., rated as 3, and items indicating less adverse experiences were rated downwards, i.e., rated as 1.items indicating common experiences were rated downwards, i.e., rated as 1.items with a broad variety of experiences implying possible dilemmas in the weighting procedure were rated as 2 to achieve a middle value.

### 2.3. Content Validation

The WICAC was based on a formative model with a theoretical approach. The validation included content validation with face validity, relevance, and comprehensiveness according to the COnsensus-based Standards for the selection of health Measurement INstruments (COSMIN) checklist [41].

#### 2.3.1. Face Validity and Relevance

The validation relied on the subjective face validity and relevance of the CLAM questionnaire. Face validity was tested subjectively by a simple overview by evaluating the questionnaire according to its length and the appropriateness of its items, followed by an evaluation of the pilot testing [28]. For WICAC, we assessed the construct again while adapting it to childhood and adolescence conditions. The appropriateness of items was evaluated in the validation of the CLAM, by which missing observations should be less than 3%, according to the guidelines of Mokkink et al. [28,41].

#### 2.3.2. Comprehensiveness

Two literature searches in PubMed were conducted with separate aims:

Search I: Defining a construct of childhood and adolescence conditions.

Search II: Determining relevant measurement instruments for childhood and adolescence adverse conditions and identifying outcomes relevant to the initial hypothesis testing (Figure 2) (Appendix A).

### 2.4. Construct Validation

Construct validation was based on hypothesis testing. Validity was assessed using the COSMIN checklist [41].

We hypothesized an increasing risk for poor health outcomes with increasing adversities, considering risk ratios (RR) for variables with adequate literature to support specific RR estimates.

#### 2.4.1. Hypotheses with Estimates

We sought to estimate potential risk ratios based on the literature regarding ACEs. We found estimates for cardiovascular disease [2,4,6,13], cancer [4,6,13], obesity [6,13], depression [2,4,6,13], anxiety [2,6,13], daily smoking [2,6,13], and alcohol addiction [2,6,13]. The estimates portray a risk at low adversity as opposed to no adversity, and severe adversity opposed to no adversity (Table 1).

#### 2.4.2. Hypotheses without Estimates

We hypothesized that an increasing risk of having any of the following outcomes was associated with an increasing WICAC score: poor health, back pain, high BMI, low vitality, smoking (grams of tobacco a day), heavy drinking, alcohol, subjective social status, low social status, and education. We further hypothesized that the association was stronger for the psychological, behavioural, and social outcomes than for the somatic outcomes. Furthermore, we hypothesized that biopsychosocial variables, such as poor health and low vitality, would show the strongest associations.

#### 2.4.3. Discriminative Validation

When evaluating construct validity, hypothesis testing is mainly focused on expected positive correlations with instruments measuring related constructs (convergent validity) as the hypotheses above [41]. However, some of the isolation of the construct may preferably contain hypotheses about what the construct of interest is not (discriminative validation) [41]. For discriminative validation, we used the measure handgrip strength and hypothesized that the association between handgrip strength and WICAC scores would be non-significant and point in either direction as we hypothesized that there would be no direct effect of ACEs on handgrip strength later in life.

### 2.5. Explanatory Measurement Variables

For our hypotheses, we categorized WICAC into five categories as shown in Table 2.

The categories make the index simplistic and interpretable as well as somewhat comparable to an unweighted cumulative effected measure, as low adversity indicates experiencing 1 moderate or 2 low adversities. Moderate adversity indicates experiencing 2 severe adversities or 3 moderate adversities. Severe adversity indicates experiencing 4 severe or 5 moderate adversities, while the category for very severe adversity indicates a high alert severity with more than 4 severe experiences. See Section 3.2 for more details on the included variables.

### 2.6. Dependent Measurement Variables

#### 2.6.1. Somatic Measurements

Cardiovascular disease and cancer were self-reported and obtained from the question: “Has a doctor ever told you that you had any of the following: heart attack/stroke, cancer”, answer yes/no.

Poor health was assessed with a single item from the 12-item Short-Form Health Survey (SF-12) [42] and obtained from a five-point Likert scale “In general, would you say your health is: 1 = Excellent, 2 = Very good, 3 = Good, 4 = Fair, 5 = Poor”. We dichotomized poor health as 1–3 = No and 4–5 = Yes.

Back pain was assessed using the Bodily Distress Syndrome (BDS) Checklist [43] with a five-point Likert scale: “During the last 12 months, have you been bothered by back pain?: 1 = Not at all, 2 = A bit, 3 = Somewhat: 4 = Quite a bit, 5 = A lot”. We dichotomized the variable to be 1–2 = no and 3–5 = yes. The central option was included as yes, as the question does not cover a chronic condition with a diagnosis but only a symptom span of 12 months.

BMI was obtained from the general examination and included both as a continuous variable with unit kg/m^2^ and a dichotomous variable for obesity according to the WHO guidelines with a cut-point for obesity class I–III with a BMI > 30 kg/m^2^.

As an estimate of muscle strength, handgrip strength was measured with a Jamar dynamometer (JAMAR^®^, pounds). Handgrip was obtained as a continuous score, calculated from the mean of two repetitions with the participant’s dominant hand. [44].

#### 2.6.2. Psychological Measurements

The variables for anxiety and depression were self-reported and obtained from the question: “Has a doctor ever told you that you have any of the following—anxiety, depression”, answer yes/no. Measurements on low vitality was assessed with a single item from SF-12 [42] on a 5-point Likert scale “How much of the time during the past 4 weeks, did you have a lot of energy? 1 = All the time, 2 = Most of the time, 3 = Some, 4 = A little of the time, 5 = None of the time”. The variable of low vitality was dichotomized as no = 1–3 and yes = 5.

#### 2.6.3. Behavioural Measurements

All data on behaviours were self-reported. Smoking was dichotomized as daily smoking present or previously vs. never been a smoker/occasional smoker. Furthermore, we obtained grams of tobacco used per day by calculating grams from the number of cigarettes, cheroots, and cigars, respectively, and grams of pipe tobacco by cases for daily present or previously smokers. This variable was continuous.

Alcohol consumption was measured as units per week, calculated from a self-reported item: “How much of the following have you consumed weekly for the past 12 months?—Regular beers, strong beers, glasses of wine, glasses of liquor, glasses of snaps/hard liquor”. All beverage consumptions were transformed into units and added up into a continuous variable. A cut-point of >35 units weekly was chosen to describe a condition of heavy drinking.

For measuring alcohol addiction, we used the CAGE measurement: “Have you ever felt you ought to cut down on your drinking?”; “Have people annoyed you by criticizing your drinking?”; “Have you ever felt bad or guilt about your drinking?”; and “Have you ever had a drink as the first thing in the morning to steady your nerves or get rid of a hangover (eye-opener)?” [45]. We used a cut-point at answering yes to >1 item in CAGE, for defining alcohol addiction [46].

#### 2.6.4. Socioeconomic Measurements

All data on socioeconomic measurements were self-reported.

Social status was measured with a subjective item on a 10-point rating scale. [47]. In addition, we dichotomized the variable with a score ≤ 3 defining “low social status”.

The variable for low education was dichotomous and obtained from the question: “Do you have vocational training beyond elementary school?” answer = yes/no [48].

### 2.7. Statistical Analysis

Stata, version 16, was used for all analyses [49].

#### 2.7.1. Descriptive Statistics

Descriptive statistics were conducted to display the main characteristics of the study sample, and comparisons of categories were performed using one-way ANOVA and χ^2^.

#### 2.7.2. Regression Analysis

Each of the biopsychosocial items was considered as an individual dependent variable. Risk ratios (RR) were estimated in 14 models with WICAC as the primary independent variable in binominal regression analyses for dichotomous dependent variables. Mean Differences (MD) for continuous dependent variables were estimated in five models with WICAC as the primary exposure in a general linear model. In all analyses, no adversity was used as a reference group for WICAC. RR and MD were chosen for easy interpretation. We inspected covariates for transformation in the binomial regression and residuals in the linear regression to assess if the models fitted the data adequately. Overall differences between the five categories no/low/moderate/severe/very severe adversity were assessed by large-sample Wald tests as a standard for testing statistical hypotheses. Both RR and MD were presented as crude and adjusted with 95% CI.

The regression analyses were conducted hierarchically in three steps: First, a crude analysis was conducted. Second, an analysis adjusted for age and sex was conducted. Third, an analysis with advanced adjustments was conducted. The advanced adjustments included: (A) cardiovascular disease and cancer adjusted for age, sex, education, self-reported social status, BMI, and smoking; (B) poor health, back pain, BMI, depression, anxiety, low vitality, daily smoking, amount of smoking, heavy drinking, and alcohol addiction adjusted for age, sex, education, and self-reported social status; and (C) education, self-reported social status and low social status adjusted for age and sex.

#### 2.7.3. Sensitivity Analysis

Three additional analyses of WICAC were included to investigate the sensitivity of the results to the choices made in the development of WICAC. Two analyses investigated the sensitivity regarding missing values, i.e., one containing participants who had completed a minimum of 50% of the items and one where all participants had completed at least 1 item (Appendix A). The third analysis investigated the sensitivity in the weighting, as we created an unweighted index (details are shown in Appendix A). All indices were categorized and analysed with binominal and linear regression similar to those involving the WICAC, with results as RR and MD. Differences were evaluated qualitatively.

## 3. Results

### 3.1. Population Characteristics

A total of 6360 participants were included with 54.6% females (Table 3), of whom 64.4% women had experienced very severe adversity. The median age was 54 years for the total sample and 49 years for participants with very severe adversity. A total of 88 non-responders were excluded.

The prevalence of all outcomes for our hypotheses increased across categories from low to very severe adversity. The significant difference in variance across index categories increased with increasing severity of adversities (Table 4). The most common poor biological outcome was back pain with a total prevalence of 26%, increasing to 37.1% for the very severe adversity category. Obesity had the second highest prevalence, with a median BMI at 25.4 kg/m^2^ for all participants. Self-reported poor health was the third most prevalent biological outcome with 1 in 5 in the very severe adversity category experiencing poor health. Cardiovascular diseases and cancer were the rarest outcomes with less than 10 individuals in the groups for very severe adversity. For the psychological outcomes, low vitality was the most common, followed by depression and anxiety. While the prevalence of anxiety was lower, anxiety had the strongest increase with a 4-fold increase in prevalence from no adversity to very severe adversity. For the behavioural outcomes, nearly half of the population were or had been daily smokers, with 58.3% in the very severe category, consuming a median of 15–17 g of tobacco a day. For alcohol variables, the most common was having an alcohol addiction. This accounted for 5.5% of the total sample; 1.7% with heavy drinking. For social outcomes, 26.4% of the population did not have any education beyond elementary school, with 38.2% in the very severe adversity category. Approximately 2% rated their social status as low while the median social rating in the total sample was 7.

### 3.2. Development of the WICAC

To develop the construct, a review of the field was carried out [33] (Appendix A). Based on items from the CLAM and the severity weighing of each adversity, the WICAC contained 29 items, including eight items with the weight of one, somewhat representing low adversity; 13 items with the weight of two, representing moderate adversity; and eight items with the weight of three, representing more severe adversity For each item it was possible to measure up to three events as participants could write three different ages (age at the time of the event). It was also possible to add an age interval for the experienced adversities across a span of years. Each time period weighted the double effect of a single event. It was therefore possible to contain a maximum score for one item at 15 if an adverse experience with a weigh of three had been experienced on three different occasions as well as for a time period, making the highest possible score 236. The descriptions and distribution of adversities can be found in Appendix A.

The items and the weight of each item can be found in Table 4. The selection of items and discussion of the weights can be found in Appendix A.

### 3.3. Validation of the WICAC

The content and construct validity was validated in accordance with the COSMIN checklist and met all the general requirements, such as the assessments of the items relevance and purpose for the measure according to content validity and a priori formulated hypotheses testing with analyses of missing data [41]. Thus, all items were theoretically evaluated as relevant for the construct to be measured. Similarly, all items were evaluated in the light of WICAC being a mainly discriminative measurement in a formative model (Figure 2).

### 3.4. Hypothesis Testing

Results are shown in Table 5, Table 6, Table 7 and Table 8. Overall, the results were in accordance with our hypotheses, as results showed a tendency towards an increase in risk with more severe adversity. For the presumed strong associations (poor health, anxiety, depression, low vitality, social status, and lower education), associations did not change markedly from the crude to the adjusted analyses. Anxiety and depression matched our hypothesized estimates when adjusting for sex, age, and social factors. The measures of low social status showed an almost 3-fold increase when having experienced very severe adversity in childhood. Cardiovascular disease and cancer showed less strong associations, which leaves our hypotheses undecided. The associations for back pain, BMI, and obesity were stronger and matched our hypotheses. Results for behavioural and social outcomes showed a slight increase in risk, with increasing adversity, except for heavy drinking, with a 4-fold risk for drinking above 35 units a week when having been exposed to very severe adversity in childhood. Handgrip strength decreased with increasing index severity.

### 3.5. Sensitivity Analysis

#### 3.5.1. Missing Values

Overall, results from the sensitivity analyses showed no prominent differences from the main analyses. This accounted both for the analysis including participants with minimum 50% completed items (N = 7371) and the analysis including participants with >1 question answered (N = 7404). The characteristics of the analysis for >1 answered had a higher age, with a median age of 54 years, and had an even distribution of sexes, as 49.9% were women. All risks for poor somatic health outcomes were similar to main analyses. All psychological and social outcomes showed a slight increase across all categories compared to main analyses, except for anxiety that showed a slight decrease in risk for poor outcome, as well for the behavioural outcome variables (Appendix A).

#### 3.5.2. Unweighted Index

Overall outcomes for the WICAC showed an increase across severity categories with a large increase in the last category, portrayed as an exponential curve, with higher risk ratios to the poor outcomes for the very severe adversity category. The unweighted index showed more tendencies to smooth out the effect in the very severe category. Compared to the WICAC, the behavioural outcomes, social status, and BMI showed a clear trend for the unweighted index, as the risk in the low adversity category increased, the moderate category decreased, the severe category increased, and the very severe category decreased (Appendix A).

Comparing the unweighted measure in the sensitivity analysis to the WICAC shows the apparent effect of a more nuanced, weighted measure, as 358 individuals are otherwise wrongly placed in the low adversity category, when they in fact have experienced more hardship. Similar results are shown for the other direction as individuals are wrongly placed in a higher adversity category, although they experienced less hardship (Appendix A).

### 3.6. Missing Analysis of Outcome Measures

Results showed an amount of missing data for a number of outcomes, especially for alcohol measures with approximately 5% missing. Participants with missing values on alcohol had a higher prevalence in the severe and very severe category, with a prevalence of 57.1% for heavy drinking in the very severe category and a prevalence of 40.9% for alcohol addiction in the very severe category. As for cardiovascular disease, missing data accounted for approximately 30% in the very severe category, while missing data on cancer was 20% for the very severe category. The lowest rate of missing was found for the outcomes obesity, poor health, back pain, low vitality, social status, and smoking, as they all had missing values <1%. More detailed descriptions can be found in Appendix A.

## 4. Discussion

### 4.1. Summary of Findings

Both content and construct validity were evaluated as acceptable based on the COSMIN checklist [41]. The results of our hypotheses testing supported our a priori formulated hypotheses, with the strongest associations between the WICAC and the psychosocial and behavioural outcomes. For cardiovascular disease and cancer, we did not find any significant associations. The hypothesis was undecided for the discriminative validity, as the results were non-significant. The sensitivity analyses supported the use of the WICAC with full data and the usage of the weighing component as the weight nuances experienced adversities. Not including that the weighing component may induce individuals being misclassified and wrongly placed in the lower adversity category when they may actually have experienced more hardship.

### 4.2. Interpretation of Findings

#### 4.2.1. Development of the WICAC

As the construct of the WICAC is based on a formative model and is a multi-item measure, it was important to include all aspects of the construct to rule out the risk of the construct not measuring comprehensively. The formative model entails that the result would be an index and not a scale, as an index contains several dimensions summarized in one score, whereas a scale is based on reflective models and unidimensionality. As we understand the construct as a multidimensional index, we aimed for the index to cover aspects of abuse, neglect, household dysfunctions, community factors, disasters, and different types of loss. While investigating the literature and discussing the surroundings of the construct, we decided to include personal injuries and illness as well. The multi-item measure makes it possible to investigate the construct in detail and to cover more of each aspect. WICAC is a retrospective measure, while the purpose of the measure is mainly discriminative. As the objective is to both investigate the consequences of childhood and adolescent conditions, it can also be used for adjustment-analysis in larger population-based studies. Furthermore, the WICAC can be included in a predictive model with other factors as well [32].

#### 4.2.2. Validation of the WICAC

When investigating the prevalence of each item in the WICAC, all items were present, except combat experience, which was only experienced by <5 individuals. Face-validity and relevance were well documented in the CLAM [28]. Comprehensiveness was well investigated through a minor review of the literature and other known measurements of childhood and adolescent adverse conditions [33]

The hypotheses testing met the design requirements from the COSMIN checklist [41]. We made a thorough sensitivity analysis on the WICAC indices with at least one item fulfilled, 50% items fulfilled and the WICAC with full data. The sensitivity analyses led us to believe that the reduced full data set was adequate for the hypotheses testing. Hypotheses for RR estimations, as well as the directions for each hypothesis, were formulated a priori, according to findings in the literature.

#### 4.2.3. Hypothesis Testing

The non-significant results for cardiovascular disease and cancer may be due to the small sample of participants in these categories. Additionally, the estimates of risk ratios for poor medical outcome from the literature were based on fewer studies, as most studies in ACEs focus on psychological outcomes, with approximately 40% more studies on psychosocial and behavioural outcomes than on biological outcomes [2]. Furthermore, people who had experienced very severe adversity in childhood had a significantly lower age at the time of inclusion. This may be due to the retrospective design in the measure as the older participants may not remember their ACEs or it may be due to an overall higher mortality. A study by Johnson et al. found a major risk for an overall mortality, increasing with a cumulative adversity with experiencing > 5 adversities, estimated hazard ratio at 1.91 CI 95% (1.25–2.32) in reference to those who had experienced 0–2 adversities [5]. The high mortality risk might be the result of the overall poorer health outcome, as well as an overrepresentation of lifetime suicide attempts in populations experiencing ACEs [12]. This also explains the small number of participants in the medical categories, as some participants may have been too young to have developed medical and somatic outcomes. We stress the need for more studies with more power in the medical and somatic fields.

#### 4.2.4. Sensitivity Analyses

Variations in the sensitivity analysis between the three indices based on the number of missing values might be due to more missing values in the alcoholic questionnaires and psychological outcomes. The sensitivity analysis for >1 completed items was not diverging from the analysis of the WICAC in this population. However, they might not be generalizable to other populations, and we can therefore not make recommendations for missing values when working with the WICAC.

The sensitivity analysis in which the weighted and unweighted indices were compared showed large variations as expected. Most outcomes showed clear tendencies of increasing across severity categories with a large increase in the last category for the weighted index, whereas the unweighted cumulative index more often showed a decrease in the very severe category. These results indicate that the weighted index is a more sensitive and precise measure as all behavioural outcomes had stronger associations for the low adversity categories compared to the unweighted index. This may be due to the inclusion of participants with an experience with the weight of 3. Furthermore, we found lower risk ratios in the moderate adversity category for the unweighted index when we relocated items with the weight of 3 from the category. We believe these results indicate that the weighted measure is more realistic, and that the weight is adequate for the WICAC. Results for lower education indicated a higher risk according to the cumulative effect and not the weighted effect, which may be due to education being a more stable and less directly associated variable. We recommend that future studies take these results into consideration.

### 4.3. Strengths and Limitations

The main strength of the development of WICAC was its comprehensiveness, both due to its theoretical approach, the inclusion of an expert panel, and the inclusion of other aspects in childhood and adolescent conditions, as opposed to other well-known measures of childhood adversities. The combination of developing a new, more comprehensive index of childhood adversity with a variety and range in variables provides a strong theoretically founded and validated index.

Another major strength was the weighting of each item, thereby respecting that each adverse experience may not have the same influence on the individual’s life across all items. By weighing the items, the WICAC fills a gap in the research field and may be a more accurate instrument than the existing measures.

Furthermore, we developed and validated the WICAC in a large random sample of the general adult population, comprising both sexes with an age range of 50 years. Although the participation rate was relatively low, the response rate of the WICAC was high (85.9%). As the WICAC is a validated retrospective measure, the time perspective removes the concern of temporality, ensuring that the exposure occurred before the outcome, as adverse events happened before the age of 18 years and our outcomes were measured as lifelong, at any time in life.

Another strength is that our validation relied on an already validated instrument, the CLAM [28].

An important limitation was the restrictions on the inclusion of items, which was limited to the items included in the CLAM, meaning that it was not possible to include variables for bullying, parental drug or alcohol abuse, protective factors or adverse experiences that had not taken place but came close, such as attempted rape, attempted suicide by close relatives or near-death experiences [50]. However, it can be argued that people who experienced bullying would have answered “yes” to the item for emotional abuse, and that aspects of alcohol or drug abuse in the household are also implicitly included in household dysfunction. This may present a risk of misclassification; however, relevance was deemed fit in the CLAM [28]. The lack of including protective measures, such as social support and coping strategies, may mediate the effect of adverse life events, as children with social support and healthy coping strategies may not meet the poor biopsychosocial health outcomes, and children without support may experience more severe health outcomes [23,30,51]. It is noticed that items in the formative model are not interchangeable, i.e., they cannot replace one another, and that they must all be included [32], and it is therefore recommended to include the above items in future studies; however, preferably after pilot testing of the WICAC to ensure the comprehensibility of the index.

Furthermore, it can be discussed if increased accuracy could be gained by including a weight linked to different age groups for each item, as some studies urge the importance of sensitive periods, i.e., by scoring differently according to age intervals [19,23,30,52]. Nevertheless, it is unclear how the weight should be evaluated, and it would therefore be preferable to perform a factor analysis or similar analysis to investigate each age group and its effect on each outcome to determine a statistically evaluated weight. This would, however, demand more power than was possible in this study.

Another limitation was the limited number of confounding variables in our hypothesis since, e.g., cohabitation status was associated with mental disorders in a non-responder analysis [53]. However, by including age, sex, education, and the broader measure of social status, we believe the models were appropriate for the validation hypotheses.

As individuals were excluded from study participation if they were not born in Denmark or were not Danish citizens, this may have induced a minor selection bias as these groups are naturally at more risk for experiencing adversities that are not common in Denmark (e.g., combat experience, bombing, natural disasters). However, this is more adequate for refugees, and not for other foreigners living in Denmark. Furthermore, it could be argued that it may not be adequate to validate adversities that are rarely experienced in Denmark in a Danish sample, thus limiting the generalizability and validity of the index. However, the purpose of this study is to be used internationally, and it was therefore decided to keep those adversities in the index, yet this limitation should be kept in mind when using the index. As for the risk of selection bias for the 88 non-responders, there were very few non-responders compared to the total population, and a non-responder analysis showed a tendency of non-responders to have a higher prevalence of mental disorders associated to cohabitation status similar to responders with a high prevalence of mental disorders. Therefore, there was no indications of the results being biased by non-responders [53].

Another important limitation is the use of self-reported data, introducing a risk of recall and information bias. However, the adverse experiences reported are not trifling, and a person should therefore be able to remember such adverse experiences even although they have not taken place in recent years. It can also be hypothesized that there is a risk of underestimating the associations, as recalling some of the adverse events may be too burdensome for some to answer. It is also a valid point for the risk of underestimation, that there are large missing percentages relative to the outcomes for alcohol consumption, alcohol abuse, cancer, cardiovascular disease, education, social status, and anxiety.

Another possible limitation for the use of self-reported data is that a clinically established diagnosis for anxiety, depression, cardiovascular disease, and cancer cannot be procured. Therefore, some of our cases might represent a false positive and controls a false negative. However, as we intended to measure the lifelong effect, we chose to use these simpler self-reported yes/no answers to determine the outcomes instead of using other standardized screening questionnaires measuring symptoms within a shorter period of time. Additionally, it can be assumed that people are able to correctly remember and answer if they have ever been diagnosed with a chronic disease such as cancer. The results are therefore likely to be valid although not representing actual clinical diagnoses. As for the measurement of subjective social status, it has been shown to give a more accurate measure than common objective measurements (e.g., income), considering the possibilities of those without education but with high income as well as students with a high education but a low income [47,54].

Another limitation is the use of only one variable to decide the discriminative validity. However, it was not possible to find other variables in our data eligible for discriminative validity, and we therefore suggest that this should be a key point for further validation studies.

### 4.4. Implications and Further Research

The main implication is the opportunity to do further and more accurate research in childhood and adolescent conditions. As shown in a large systematic review and meta-analysis from 2019, the inequality and inequity in health associated with ACEs are a major cost at an estimated USD 581 billion in Europe, where approximately 75% of these costs came from individuals experiencing more than two adverse experiences in childhood [6]. By investigating these adversities in specific for childhood and adolescence, it may be possible to argue for early recognition and new policies to prevent the cumulative effect of experiencing adversity in childhood. To achieve this, we recognize the need for the further development of the index, as well as a need for longitudinal studies with more power to investigate the biopsychosocial consequences of severe adversities measured by the WICAC. Furthermore, it would be preferable to integrate epidemiological studies into adverse childhood conditions, with for example immunological measures, epigenetics, and neurology for a deeper understanding on how ACEs affect our health.

## 5. Conclusions

The WICAC is an improvement in the field as it includes a weighing component and an extensive range of adverse childhood experiences (i.e., physical, sexual, and emotional abuse; neglect; household dysfunction; community factors; disasters; bereavement, loss and injures; own injures and health conditions). Furthermore, it includes age at exposure, and a time period if the adversity happened over a span of several years.

The WICAC met the criteria for the COSMIN checklist, including construct and construct validation with hypothesis testing. In conclusion, the WICAC is an adequate measurement of childhood and adolescent adversities in population-based studies.

## Figures and Tables

**Figure 1 ijerph-19-13251-f001:**
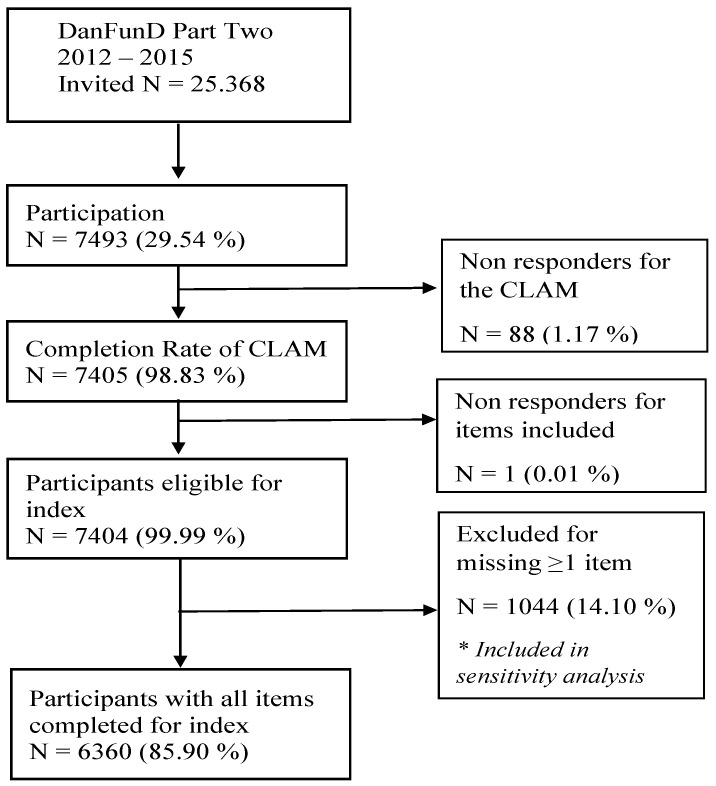
Flow Chart of participation.

**Figure 2 ijerph-19-13251-f002:**
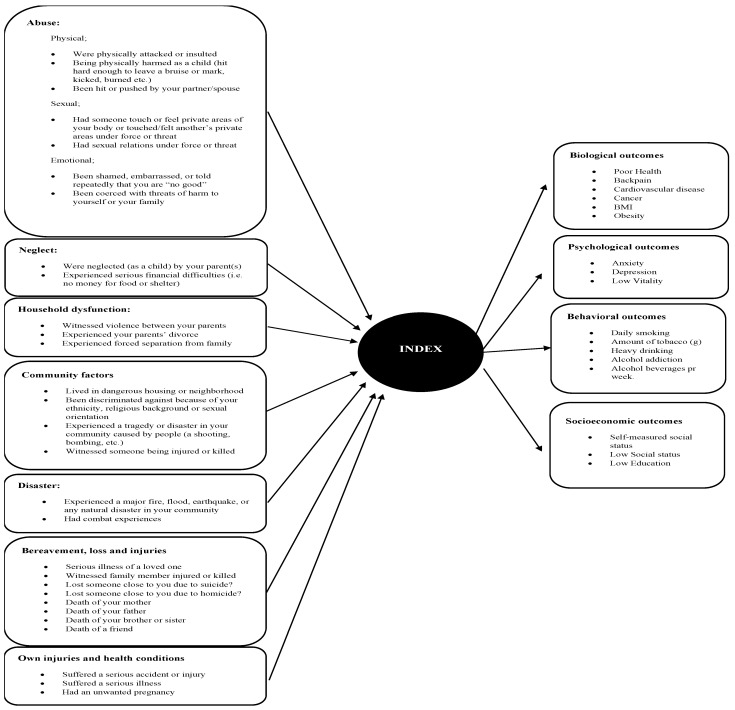
The WICAC showing the items forming the index. On the right side of the model, it shows the biopsychosocial outcomes from our hypotheses.

**Table 1 ijerph-19-13251-t001:** Hypothesis estimates based on results from studies using ACE Q scores.

Outcome	References, CI		Estimates, RR	
	1 ACE	>4 ACE	Low Adversity	Severe Adversity
Cardiovascular disease	0.95–1.24	0.91–2.06	1	2.5–3.0
Cancer	0.83–1.63	0.57–2.20	1	2.0–2.5
Obesity	0.99–1.42	0.85–2.34	1	1.2
Depression	1.51–1.72	2.14–5.03	1.5	2.5–3.0
Anxiety	1.17–1.77	2.19–2.98	1.2–1.4	2.5–3.0
Daily Smoking	1.19–1.38	1.71–2.18	1.2	1.9–2.1
Alcohol Addiction	1.22–1.87	1.13–3.95	1.2–1.5	1.8–2.1

CI resembles the lowest and highest CI from the included studies in this analysis. Abbreviations: CI = Confidence Interval; RR = Risk Ratio. Low adversity equals 1 adversity on the ACE score. Severe adversities equal ≥ 4 adversities on the ACE score.

**Table 2 ijerph-19-13251-t002:** WICAC Categories.

Category	Sum Score (0–236)
No adversity	0
Low Adversity	1–2
Moderate Adversity	3–7
Severe Adversity	8–13
Very Severe Adversity	>13

**Table 3 ijerph-19-13251-t003:** Population Characteristics divided into index categories.

Variable	Total WICAC (N = 6360)	No Adversity (N = 3383)	Low Adversity (N = 1293)	Moderate Adversity (N = 1261)	Severe Adversity (291)	Very Severe Adversity (N = 132)
Age at baseline * Median, (IQR)	54 (44–63)	55 (46–64)	49 (39–58)	51 (40–61)	50 (2137–58)	49 (35–57)
Sex, Female% (N)	54.6 (3472)	54.8 (1853)	53.8 (696)	53.5 (675)	56.0 (163)	64.4 (85)
Cardiovascular disease% (N)	3.6 (228)	3.9 (130)	3.0 (39)	3.2 (40)	5.3 (15)	3.1 (<5)
Cancer% (N)	6.9 (438)	7.5 (248)	5.6 (72)	7.3 (91)	6.7 (19)	6.2 (8)
Poor Health * % (N)	8.6 (546)	7.4 (250)	6.8 (87)	10.7 (134)	17.3 (50)	18.9 (25)
Back pain * % (N)	26.0 (1651)	24.2 (813)	26.5 (340)	28.2 (353)	33.2 (96)	37.1 (49)
BMI Median (IQR)	25.4 (22.9–28.5)	25.3 (22.8–28.4)	24.9 (22.6–28)	25.6 (22.8–28.55)	25.1 (23.6–28.5)	25.3 (22.35–29.05)
Obesity % (N)	17.0 (1078)	16.7 (564)	15.6 (202)	18.3 (230)	18.6 (54)	18.2 (24)
Anxiety * % (N)	4.8 (302)	3.9 (130)	4.5 (58)	5.2 (65)	10.2 (29)	15.4 (20)
Depression * % (N)	12.0 (765)	10.0 (332)	12.4 (159)	13.6 (170)	21.7 (62)	32.3 (42)
Low vitality * % (N)	17.7 (1128)	16.4 (549)	16.4 (210)	19.4 (243)	27.5 (79)	35.6 (47)
Subjective Social Status * Median (IQR)	7 (6–8)	7 (6–8)	7 (6–8)	7 (6–8)	6 (5–7)	6 (5–7)
Low social status * % (N)	2.4 (152)	1.6 (54)	2.6 (33)	3.6 (45)	4.2 (12)	6.1 (8)
Low Education * % (N)	26.4 (1971)	23.2 (783)	29.1 (375)	31.4 (395)	30.7 (89)	38.2 (50)
Daily smoking * % (N)	47.9 (3049)	46.2 (1560)	46.1 (594)	52.2 (657)	55.5 (161)	58.3 (77)
Smoking g. Median (IQR)	15 (10–20)	15 (10–20)	15 (10–20)	15 (10–20)	15 (10–20)	17 (12–20)
Heavy drinking % (N)	1.7 (107)	1.8 (57)	1.4 (17)	1.9 (22)	1.9 (5)	4.8 (6)
Alcohol, Addiction % (N)	5.5 (351)	5.4 (175)	5.3 (65)	6.2 (73)	9.4 (25)	10.8 (13)
Alcohol, Units a week Median (IQR)	5 (2–10)	5.5 (2–10.5)	4 (1.25–9)	5 (2–10)	4 (2–10)	3 (1–10)

Obesity measured at >30 kg/m^2^. Low social status is measured as a subjective social status < 4. Heavy drinking is measured as >35 units pr. week. Addiction is measured as >1 on the CAGE score. * Significant at *p* < 0.00001 in chi^2^.

**Table 4 ijerph-19-13251-t004:** Weighted Index for Childhood Adverse Conditions (WICAC).

Categories Items	Possible Score *		
	Per Event	Periode	Max.
**Index**	**1–3**	**2–6**	**236**
**Abuse, Physical**			
Was physically attacked or insulted	1	2	5
Being physically harmed as a child (hit hard enough to leave a bruise or mark, kicked, burned, etc.)	2	4	10
Been hit or pushed by your partner/spouse	2	4	10
**Abuse, Sexual**			
Had someone touch or feel private areas of your body or touched/felt another’s private areas under force or threat	3	6	15
Had sexual relations under force or threat	3	6	15
**Abuse, Emotional**			
Been shamed, embarrassed, or told repeatedly that you are “no good”	2	4	10
Been coerced with threats of harm to yourself or your family	2	4	10
**Neglect**			
Was neglected (as a child) by your parent(s)	2	4	10
Experienced serious financial difficulties (i.e., no money for food or shelter)	1	2	5
**Household dysfunction**			
Witnessed violence between your parents	2	4	10
Experienced your parents’ divorce	1	-	3
Experienced forced separation from family	2	4	10
**Community Factors**			
Lived in dangerous housing or neighborhood	1	2	5
Been discriminated against because of your ethnicity, religious background, or sexual orientation	1	2	5
Experienced a tragedy or disaster in your community caused by people (a shooting, bombing, etc.)	1	2	5
Witnessed someone being injured or killed	2	-	6
**Disaster**			
Experienced a major fire, flood, earthquake, or any natural disaster in your community	2	-	6
Had combat experiences	3	6	15
**Bereavement, loss and injuries**			
Serious illness of a loved one	1	2	5
Witnessed family member injured or killed	2		6
Lost someone close to you due to suicide	3		9
Lost someone close to you due to homicide	3		9
Death of your mother	3		9
Death of your father	3		9
Death of your brother or sister	3		9
Death of a friend	2		6
**Own injuries and health conditions**			
Had an unwanted pregnancy	1		3
Suffered a serious accident or injury	2	-	6
Suffered a serious illness	2	4	10

* Overview of weights per item: events (up to 3), periods, and total scores.

**Table 5 ijerph-19-13251-t005:** Risk Ratio and Mean Differences between WICAC and biological outcomes.

Biological Outcomes Adversity Categories	Crude			Adjusted ^1^			Adjusted ^2^		
	RR/ MD	95% CI	*p*-Value	RR/ MD	95% CI	*p*-Value	RR/ MD	95%CI	*p*-Value
Cardiovascular Disease *, RR *p* = 0.3852 Crude N = 6284, Adjusted^1^ N = 6284, Adjusted^2^ N = 6104									
Low	0.78	(0.55–1.11)	0.166	1.12	(0.79–1.59)	0.521	1.06	(0.75–1.51)	0.727
Moderate	0.82	(0.58–1.16)	0.262	1.01	(0.72–1.42)	0.962	0.90	(0.64–1.27)	0.545
Severe	1.35	(0.80–2.27)	0.259	2.06	(1.23–3.46)	0.006	1.65	(0.89–2.68)	0.072
Very Severe	0.79	(0.30–2.10)	0.636	1.23	(0.48–3.18)	0.668	1.09	(0.42–2.82)	0.864
Cancer *, RR *p* = 0.5344 Crude N = 6269, Adjusted^1^ N = 6269, Adjusted^2^ N = 6089									
Low	0.76	(0.59–0.97)	0.031	1.02	(0.80–1.32)	0.851	1.05	(0.81–1.36)	0.714
Moderate	0.98	(0.78–1.23)	0.854	1.19	(0.95–1.49)	0.132	1.20	(0.95–1.52)	0.116
Severe	0.90	(0.57–1.41)	0.639	1.26	(0.81–1.96)	0.309	1.23	(0.77–1.98)	0.383
Very Severe	0.83	(0.42–1.63)	0.582	1.11	(0.57–2.14)	0.762	0.88	(0.41–1.91)	0.745
Poor Health, RR, *p* < 0.0001 Crude N = 6331, Adjusted^1^ N = 6331, Adjusted^2^ N = 6153									
Low	0.91	(0.72–1.15)	0.426	1.02	(0.81–1.30)	0.841	1.06	(0.84–1.34)	0.613
Moderate	1.44	(1.18–1.76)	<0.0001	1.55	(1.27–1.90)	<0.0001	1.44	(1.20–1.72)	<0.0001
Severe	2.33	(1.76–3.08)	<0.0001	2.64	(2.00–3.49)	<0.0001	2.46	(1.97–3.08)	<0.0001
Very Severe	2.55	(1.76–3.70)	<0.0001	2.83	(1.95–4.10)	<0.0001	2.16	(1.83–2.91)	<0.0001
Back Pain, RR *p* < 0.0016 Crude N = 6309, Adjusted^1^ N = 6309, Adjusted^2^ N = 6119									
Low	1.09	(0.98–1.22)	0.105	1.12	(1.00–1.25)	0.041	1.13	(1.01–1.26)	0.036
Moderate	1.16	(1.05–1.30)	0.005	1.18	(1.06–1.32)	0.002	1.14	(1.02–1.27)	0.020
Severe	1.37	(1.15–1.63)	<0.0001	1.40	(1.17–1.67)	<0.0001	1.33	(1.11–1.58)	0.002
Very Severe	1.53	(1.22–1.93)	<0.0001	1.54	(1.22–1.94)	<0.0001	1.45	(1.19–1.77)	>0.001
Obesity, RR *p* = 0.3305 Crude N = 6356, Adjusted^1^ N = 6356, Adjusted^2^ N = 6164									
Low	0.94	(0.81–1.08)	0.381	1.07	(0.92–1.24)	0.400	1.06	(0.91–1.23)	0.444
Moderate	1.09	(0.95–1.26)	0.206	1.20	(1.04–1.37)	0.011	1.16	(1.00–1.33)	0.043
Severe	1.11	(0.86–1.43)	0.409	1.28	(1.00–1.65)	0.051	1.14	(0.87–1.49)	0.347
Very Severe	1.09	(0.75–1.58)	0.649	1.27	(0.87–1.81)	0.224	1.13	(0.77–1.66)	0.529
BMI; kg/m^2^, MD *p* = 0.0002 Crude N = 6356, Adjusted^1^ = 6356, Adjusted^2^ N = 6164									
Low	−0.329	(−0.327–(0.031))	0.030	0.121	(−0.172–(−0.415))	0.417	0.121	(−0.176–(−0.417))	0.425
Moderate	0.219	(−0.081–0.520)	0.153	0.554	(0.260–0.848)	<0.0001	0.506	(0.209–0.803)	0.001
Severe	0.463	(−0.093–1.019)	0.103	0.978	(0.437–1.520)	<0.0001	0.765	(0.210–1.320)	0.007
Very Severe	0.317	(−0.491–1.124)	0.442	0.944	(0.159–1.729)	0.018	0.628	(−0.167–1.424)	0.122
Handgrip strength, Discriminative Validation, MD *p* = 0.2594 Crude N = 6333, Adjusted^1^ N = 6333, Adjusted^2^ N = 6140									
Low	3.106	(1.523–4.689)	<0.0001	0.059	(−0.883–1.003)	0.901	0.081	(−0.863–1.025)	0.866
Moderate	1.529	(−0.067–3.126)	0.060	−0.852	(−1.796–0.092)	0.077	−0.458	(−1.402–0.486)	0.342
Severe	−0.021	(−3.173–2.747)	0.888	−2.526	(−4.269–(−0.783))	0.005	−1.282	(−3.048–0.485)	0.155
Very Severe	−3.358	(−7.658–0.942)	0.126	−2.671	(−5.197–(−0.144)	0.038	−2.108	(−4.643–0.428)	0.103

Adjusted^1^: Age at inclusion, sex. Adjusted^2^: Age at inclusion, sex, social status, and education; * Further adjusted for smoking, BMI in Adjusted^2^. Abbreviations: RR: Risk Ratio; MD: Mean Difference. All RR and MD are relative to the reference group = No adversity.

**Table 6 ijerph-19-13251-t006:** Risk Ratio between WICAC and psychological outcomes.

Psychological Outcomes Adversity Categories	Crude			Adjusted^1^			Adjusted^2^		
	RR	95% CI	*p*-Value	RR	95% CI	*p*-Value	RR	95% CI	*p*-Value
Anxiety, RR *p* < 0.00001 Crude N = 6278, Adjusted^1^ N = 6278, Adjusted^2^ N = 6103									
Low	1.16	(0.86–1.57)	0.338	1.12	(0.83–1.53)	0.454	1.16	(0.85–1.56)	0.347
Moderate	1.33	(1.00–1.78)	0.053	1.31	(0.98–1.75)	0.073	1.15	(0.86–1.54)	0.353
Severe	2.62	(1.78–3.84)	<0.0001	2.52	(1.71–3.71)	<0.0001	2.49	(1.70–3.66)	<0.0001
Very Severe	3.94	(2.55–6.10)	<0.0001	3.64	(2.35–5.65)	<0.0001	3.32	(2.32–4.74)	<0.0001
Depression, RR *p* < 0.00001 Crude N = 6285, Adjusted^1^ N= 6285, Adjusted^2^ N = 6109									
Low	1.25	(1.04–1.49)	0.015	1.24	(1.04–1.48)	0.018	1.23	(1.03–1.47)	0.021
Moderate	1.36	(1.15–1.62)	<0.0001	1.37	(1.15–1.63)	<0.0001	1.28	(1.07–1.52)	0.005
Severe	2.18	(1.71–2.78)	<0.0001	2.16	(1.69–2.75)	<0.0001	1.95	(1.52–2.50)	<0.0001
Very Severe	3.25	(2.48–4.25)	<0.0001	3.05	(2.33–3.99)	<0.0001	2.49	(1.97–3.13)	<0.0001
Low Vitality, RR *p* < 0.00001 Crude N = 6313, Adjusted^1^ N = 6313, Adjusted^2^ N = 6136									
Low	1.00	(0.86–1.16)	0.999	0.95	(0.82–1.10)	0.482	0.94	(0.81–1.09)	0.440
Moderate	1.19	(1.03–1.36)	0.014	1.13	(0.99–1.30)	0.071	1.06	(0.93–1.22)	0.381
Severe	1.68	(1.37–2.06)	<0.0001	1.57	(1.28–1.92)	<0.0001	1.40	(1.14–1.71)	0.001
Very Severe	2.18	(1.71–2.77)	<0.0001	2.01	(1.58–2.56)	<0.0001	1.75	(1.40–2.17)	<0.0001

Adjusted^1^: Age at inclusion, sex. Adjusted^2^: Age at inclusion, sex, social status, education. Abbreviations: RR: Risk Ratio. RR is relative to the reference group = No adversity.

**Table 7 ijerph-19-13251-t007:** Risk Ratio and Mean Differences between WICAC and behavioral outcomes.

Behavioral Outcomes Adversity Categories	Crude			Adjusted^1^			Adjusted^2^		
	RR/MD	95% CI	*p*-Value	RR/MD	95% CI	*p*-Value	RR/MD	95% CI	*p*-Value
Daily Smoking, RR *p* < 0.00001 Crude N = 6343, Adjusted^1^ N = 6343, Adjusted^2^ N = 6165									
Low	1.00	(0.93–1.07)	0.925	1.09	(1.12–1.27)	0.014	1.08	(1.01–1.16)	0.026
Moderate	1.13	(1.06–1.20)	<0.0001	1.19	(1.12–1.28)	<0.0001	1.18	(1.10–1.25)	<0.0001
Severe	1.20	(1.08–1.34)	0.001	1.31	(1.18–1.45)	<0.0001	1.28	(1.15–2.42)	<0.0001
Very Severe	1.26	(1.09–1.46)	0.002	1.36	(1.18–1.57)	<0.0001	1.31	(1.13–1.51)	<0.0001
Heavy Drinking, RR *p* = 0.0129 Crude N = 6041, Adjusted^1^ N = 6041, Adjusted^2^ N = 5872									
Low	0.79	(0.46–1.35)	0.384	1.04	(0.61–1.77)	0.895	1.11	(0.64–1.89)	0.716
Moderate	1.06	(0.65–1.73)	0.816	1.20	(0.74–1.94)	0.469	1.11	(0.68–1.83)	0.667
Severe	1.07	(0.43–2.64)	0.885	1.56	(0.63–3.83)	0.336	1.57	(0.64–3.85)	0.325
Very Severe	2.75	(1.21–6.26)	0.016	4.26	(1.95–9.34)	<0.0001	4.09	(1.85–9.04)	<0.0001
Alcohol Addiction, RR *p* = 0.0263 Crude N = 6031, Adjusted^1^ N = 6031, Adjusted^2^ N = 5863									
Low	0.98	(0.74–1.29)	0.892	0.93	(0.70–1.23)	0.597	0.93	(0.70–1.24)	0.613
Moderate	1.15	(0.88–1.49)	0.314	1.09	(0.83–1.42)	0.530	1.03	(0.78–1.35)	0.847
Severe	1.75	(1.17–2.60)	0.006	1.70	(1.14–2.54)	0.009	1.64	(1.10–2.46)	0.016
Very Severe	1.96	(1.15–3.34)	0.014	2.00	(1.18–3.39)	0.010	1.82	(1.05–3.16)	0.032
Smoking, Amount, MD *p* < 0.00001 Crude N = 3049, Adjusted^1^ N = 3049, Adjusted^2^ N = 2959									
Low	−0.064	(−0.994–0.866)	0.892	0.548	(−0.371–1.466)	0.243	0.247	(−0.680–1.175)	0.601
Moderate	1.777	(0.880–2.674)	<0.0001	2.047	(1.166–2.928)	<0.0001	1.904	(1.014–2.793)	<0.0001
Severe	1.915	(0.319–3.512)	0.019	2.503	(0.937–4.068)	0.002	2.373	(0.768–3.978)	0.004
Very Severe	2.765	(0.513–5.017)	0.016	3.759	(1.560–5.958)	0.001	3.153	(0.896–5.409)	0.006
**Alcohol Consumption, MD *p* = 0.1320** **Crude N = 6041, Adjusted^1^ N = 6041, Adjusted^2^ N = 5872**									
Low	−1.235	(−1.817–(−0.653))	<0.0001	−0.170	(−0.717–0.377)	0.542	−0.124	(−0.680–0.432)	0.663
Moderate	−0.410	(−1.000–0.180)	0.173	0.298	(−0.253–0.848)	0.289	0.309	(−0.250–0.868)	0.278
Severe	−0.869	(−1.976–0.238)	0.124	0.474	(−0.556–1.503)	0.367	0.622	(−0.437–1.680)	0.250
Very Severe	−0.159	(−1.747–1.429)	0.845	1.498	(0.025–2.458)	0.046	1.606	(0.103–3.109)	0.036

Adjusted^1^: Age at inclusion, sex. Adjusted^2^: Age at inclusion, sex, social status, education. Abbreviations: RR: Risk Ratio; MD: Mean Difference. All RR and MD are relative to the reference group = No adversity.

**Table 8 ijerph-19-13251-t008:** Risk Ratio and Mean Differences between WICAC and social outcomes.

Social Outcomes Adversity Categories	Crude			Adjusted^1^		
	RR/MD	95% CI	*p*-Value	RR/MD	95% CI	*p*-Value
Low Social Status, RR *p* = 0.0056 Crude N = 6298, Adjusted^1^ N = 6298						
Low	1.59	(1.04–2.45)	0.033	1.24	(0.80–1.91)	0.337
Moderate	2.23	(1.51–3.29)	<0.0001	1.80	(1.21–2.68)	0.003
Severe	2.59	(1.40–4.78)	0.002	1.92	(1.03–3.56)	0.039
Very Severe	3.75	(1.82–7.73)	<0.0001	2.79	(1.36–5.72)	0.005
Low Education, RR *p* = 0.0064 Crude N = 6342, Adjusted^1^ N = 6342						
Low	1.25	(1.13–1.39)	<0.0001	1.09	(0.98–1.20)	0.097
Moderate	1.35	(1.22–1.50)	<0.0001	1.19	(1.07–1.31)	0.001
Severe	1.32	(1.10–1.59)	0.003	1.12	(0.94–1.33)	0.191
Very Severe	1.34	(1.31–2.06)	<0.0001	1.27	(1.03–1.57)	0.024
Social Status, MD *p* < 0.00001 Crude N = 6297, Adjusted^1^ N = 6297						
Low	−0.0633	(−0.152–0.026)	0.164	−0.019	(−0.109–0.071)	0.680
Moderate	−0.243	(−333–(−0.153))	<0.001	−0.211	(−0.301–(−0.121)	<0.0001
Severe	−0.409	(−0.576–(−0.243))	<0.001	−0.353	(−0.520–(−0.186)	<0.0001
Very Severe	−0.558	(−0.799–(−0.317))	<0.001	−0.481	(−0.721–(−0.241)	<0.0001

Adjusted^1^: Age at inclusion, sex. Abbreviations: RR: Risk Ratio; MD: Mean Difference. All RR and MD are relative to the reference group = No adversity.

## Data Availability

Data are not available online as the research is still ongoing.

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
