# Peer review of "Development and Validation of the Weighted Index for Childhood Adverse Conditions (WICAC)"

_ijerph, 2022, doi:10.3390/ijerph192013251_

Round 1
Reviewer 1 Report
The researchers are to be commended for undertaking this study. With an aim to develop and validate an instrument that accounts for the broad inclusion of childhood and adolescent adverse experiences as well as recognizing the differential impact of adversity related to type and accumulation addresses a gap in the existing measures. This study is timely and relevant the readers of this journal and to health care providers in general. There are a number of strengths including the methodological approach and the associated transparency in describing the methods and methodology. The level of detailed and comprehensive description provides the reader the ability to understand and evaluate the soundness of the study and any inaccuracies.
The review of the literature, with the exception of seminal and theoretical perspectives is relevant, comprehensive, and primarily current to the past 5 years.
Rich detail explain each step of the process from literature search, to study design, hypothesis testing, results, limitations, steps for future research are provided.
The manuscript would benefit from a final editing to correct minor errors, e.g. p.15 line 454--sentence beings with a reference; p 15 lines466-468 not written grammatically correct
Author Response
The researchers are to be commended for undertaking this study. With an aim to develop and validate an instrument that accounts for the broad inclusion of childhood and adolescent adverse experiences as well as recognizing the differential impact of adversity related to type and accumulation addresses a gap in the existing measures. This study is timely and relevant the readers of this journal and to health care providers in general. There are a number of strengths including the methodological approach and the associated transparency in describing the methods and methodology. The level of detailed and comprehensive description provides the reader the ability to understand and evaluate the soundness of the study and any inaccuracies.
- Thank you for your well sought comments on this study’s relevance and methodological approach.
The review of the literature, with the exception of seminal and theoretical perspectives is relevant, comprehensive, and primarily current to the past 5 years.
- Many thanks for this accurate comment.
Rich detail explain each step of the process from literature search, to study design, hypothesis testing, results, limitations, steps for future research are provided.
- Thank you very much, we found it very important that the study showed much transparency to raise the quality of the development and validation of WICAC.
The manuscript would benefit from a final editing to correct minor errors, e.g. p.15 line 454--sentence beings with a reference; p 15 lines466-468 not written grammatically correct)
- Thank you for these important points, the paper has now been reread and edited. The sections have now been edited as:
“A study by Johnson et al. found a major risk for an overall mortality, increasing with a cumulative adversity with experiencing > 5 adversities, estimated a hazard ratio at 1.91 CI 95% (1.25-2.32) in reference to those who had experienced 0-2 adversities [5].” (Line 496-499)
“The sensitivity analysis for >1 completed items was not diverging from the analysis of the WICAC in this population. However, they might not be generalizable to other populations, and we can therefore not make recommendations for missing values when working with the WICAC.” (Line 507-510)

Reviewer 2 Report
Thank you for the opportunity to review this manuscript. It is clear how much work went into the project and the comprehensive process for validation of the measure.
Specific comments are provided in the attached PDF. There are a few areas for clarification on key components of the manuscript. The manuscript would benefit from copy editing to check for minor errors.
This research is interesting and worthwhile to the field to better understand childhood and adolescent adversity.

Author Response
Thank you for the opportunity to review this manuscript. It is clear how much work went into the project and the comprehensive process for validation of the measure.
Specific comments are provided in the attached PDF. There are a few areas for clarification on key components of the manuscript. The manuscript would benefit from copy editing to check for minor errors.
- Thank you very much for your thorough work! I have edited and included your comments in the manuscript, and listed answers to your comments here:
Lines 130-131: It is a little unclear as written the WICAC items are only questions from the CLAM? If additional questions were added, would pilot testing then be warranted?
- If additional questions were added, it would be warranted for pilot testing. This issue has been addressed in the discussion session now:
“It is noticed that items in the formative model are not interchangeable, i.e., they cannot replace one another, and that they must all be included [32], and it is therefore recommended to include the above items in future studies; however, preferably after pilot testing of the WICAC to ensure the comprehensibility of the index.” (Line 559-564)
Table 1: Were confidence intervals for any of the RR relationships provided that could be included here?
- I have edited Table 1 and included CI references, by including the lowest and highest CI intervals for the included studies. I hope this is fulfilling.
|
|
Table 1: Hypothesis estimates based on results from studies using ACE-Q scores |
|||
|
Outcome |
References, CI |
Estimates, RR |
||
|
1 ACE |
>4 ACE |
Low Adversity |
Severe Adversity |
|
|
Cardiovascular disease |
0.95-1.24 |
0.91-2.06 |
1 |
2.5-3.0 |
|
Cancer |
0.83-1.63 |
0.57-2.20 |
1 |
2.0-2.5 |
|
Obesity |
0.99-1.42 |
0.85-2.34 |
1 |
1.2 |
|
Depression |
1.51-1.72 |
2.14-5.03 |
1.5 |
2.5-3.0 |
|
Anxiety |
1.17-1.77 |
2.19-2.98 |
1.2-1.4 |
2.5-3.0 |
|
Daily Smoking |
1.19-1.38 |
1.71-2.18 |
1.2 |
1.9-2.1 |
|
Alcohol Addiction |
1.22-1.87 |
1.13-3.95 |
1.2-1.5 |
1.8-2.1 |
|
Table1: CI-intervals resembles the lowest and highest CI from the included studies in this analysis. Abbreviations, CI=Confidence interval, RR=Risk ratios. Low adversity equal 1 adversity on the ACE-score. Severe adversity equal ≥ 4 adversities on the ACE-score. |
||||
Section 2.4.3. Discriminative validation: This seems a bit randomly included. Hand grip strength is not mentioned in the introduction as an outcome regularly associated with ACEs and is not mentioned as a measure that was collected as part of the DanFunD (though it might be collected here), so this seems pretty out of place.
- Thank you for this comment. We have now clarified the term of discriminative validation in the manuscript. Handgrip strength is not mentioned in the introduction, as it is not relevant, as it is solely used for the discriminative validity. The paragraph now reads:
“When evaluating construct validity, hypothesis testing is mainly focused on expected positive correlations with instruments measuring related constructs (convergent validity) as the hypotheses above [41]. However, some of the isolation of the construct may preferably contain hypotheses about what the construct of interest is not (discriminative validation) [41]. For discriminative validation, we used the measure handgrip strength and hypothesized that the association between handgrip strength and WICAC scores would be non-significant and point in either direction, as we hypothesized that there would be no direct effect of ACE on handgrip strength later in life.” (Line 224-232)
Section 2.5: Explanatory measurement variables: It would be useful to see what is considered a low, moderate, or severe adversity. Providing an example of each might be useful here to provide context. So the sum score could range from 0-236? This is not fully clear as written.
- Thank you for this point. As the WICAC both figures as an explanatory variable and a result in this paper, the structure has been somewhat difficult, to ensure all relevant information in the method section without presenting the results. I have included a sentence referring to the result section for further detail, as well as including a sentence of the sum score in the Results section. I hope this finds you well.
“The categories make the index simplistic and interpretable, and somewhat comparable to an unweighted cumulative effected measure, as low adversity indicates experiencing 1 moderate or 2 low adversities. Moderate adversity indicates experiencing 2 severe adversities, or 3 moderate adversities. Severe adversity indicates experiencing 4 severe or 5 moderate adversities, while the category for very severe adversity indicates a high alert severity with more than 4 severe experiences. See Section 3.2 and Table 4 for more details on the included variables.” (Line 243-244)
“Each time period weighed the double effect of a single event. It was therefore possible to contain a maximum score for one item at 15, if an adverse experience with a weigh of three, had been experienced on three different occasions as well as for a time period, making the highest possible score 236.“ (Line 368-372)
Section 2.7.2: I wonder a little bit about the use of relative risk instead of odds ratios here. It would be good to have that further explained in the methodology
- RR were chosen for easy interpretation. The manuscript now reads:
"Each of the biopsychosocial items was considered as an individual dependent variable. Risk ratios (RR) were estimated in 14 models with the WICAC as the primary independent variable in binominal regression analyses for dichotomous dependent variables. Mean Differences (MD) for continuous dependent variables were estimated in five models with the WICAC as the primary exposure in a general linear model. In all analyses, no adversity was used as reference group for the WICAC. RR and MD were chosen for easy interpretation. We inspected covariates for transformation in the binomial regression and residuals in the linear regression to assess if the models fitted the data adequately." (Line 303-311)
Section 3.2: Development of the WICAC: with 1 being low adversity, 2 being moderate adversity, and 3 being severe adversity?
- Both yes and no, as the categories used in the hypotheses may include different types of adversity due the weight, as explained in the above section. I have included the following in the manuscript
“Based on items from the CLAM and the severity weighing of each adversity, the WICAC contained 29 items, including eight items with the weight of one, somewhat representing low adversity; 13 items with the weight of two, representing moderate adversity; and eight items with the weight of three, representing more severe adversity” (Line 362-365)
Line 498-500: I would be hesitant to fully say causal here, but potentially that the time perspective removes the concern of temporality, ensuring that the exposure occurred before the outcome.
- Thank you for this important point, the manuscript now reads:
“As the WICAC is a validated retrospective measure, the time perspective removes the concern of temporality, ensuring that the exposure occurred before the outcome, as adverse events happened before the age of 18 year, and our outcomes were measured lifelong, at any time in life.”
This research is interesting and worthwhile to the field to better understand childhood and adolescent adversity.
- Thank you very much for this observation

Reviewer 3 Report
The authors aim to develop and validate the instrument Weighted Index for Childhood Adverse Conditions. I believe this work is potentially important and could clearly lead to a method to comprehend the complexity and variations in severity of adverse events.
I have the following comments:
The authors analyze the data with binomial and linear regression models, however it is not clear why authors utilize these methods in comparison to other methods which are robust against both linear and nonlinear data.
It is not clear why authors utilized Wald test to calculate the test statistic. Please describe the reasons for utilizing this method in the manuscript.
Please proof read the article for grammar and punctuation
Author Response
The authors aim to develop and validate the instrument Weighted Index for Childhood Adverse Conditions. I believe this work is potentially important and could clearly lead to a method to comprehend the complexity and variations in severity of adverse events.
- Thank you very much for this comment, we believe this could improve the field.
The authors analyze the data with binomial and linear regression models, however it is not clear why authors utilize these methods in comparison to other methods which are robust against both linear and nonlinear data.
- Thank you for your point. The chosen models supply an easy interpretation of the results. Further, we validated the models by inspecting the residuals in the linear regression analyses, as well as inspecting covariates for possible transformation in the binomial regressions analyses. We have included these aspects in the manuscript, and hope that this is fulfilling for the choice of method:
"Each of the biopsychosocial items was considered as an individual dependent variable. Risk ratios (RR) were estimated in 14 models with the WICAC as the primary independent variable in binominal regression analyses for dichotomous dependent variables. Mean Differences (MD) for continuous dependent variables were estimated in five models with the WICAC as the primary exposure in a general linear model. In all analyses, no adversity was used as reference group for the WICAC. RR and MD were chosen for easy interpretation. We inspected covariates for transformation in the binomial regression and residuals in the linear regression to assess if the models fitted the data adequately." (Line 303-311)
It is not clear why authors utilized Wald test to calculate the test statistic. Please describe the reasons for utilizing this method in the manuscript.
- We chose to utilize the Wald test, as it is standard for testing statistical hypotheses. Further, we do not expect the result to differ if we had chosen likelihood ratio test or bootstrap test, as we have a large population sample. We also strive to focus on estimated effects and not so much on statistical significance and p-values (as recommended by the American Statistical Association in their statement on p-values). We have included the following in the manuscript:
"Overall differences between the five categories no/low/moderate/severe/very severe adversity were assessed by large sample Wald tests as a standard for testing statistical hypotheses. Both RR and MD were presented as crude and adjusted with 95 % CI." (Line 311-314)
Please proof read the article for grammar and punctuation
- Thank you for this valid point, the article has now been proof read.

Round 2
Reviewer 3 Report
Authors have addressed my previous comments.